# Conflicting effects of recombination on the evolvability and robustness in neutrally evolving populations

Alexander Klug[1,2¤], Joachim Krug[1]*

**1** Institute for Biological Physics, University of Cologne, Köln, Germany, **2** Institute of Integrative Biology, ETH Zurich, Zurich, Switzerland

¤ Current address: Institute of Integrative Biology, ETH Zurich, Zurich, Switzerland
* jkrug@uni-koeln.de

**Data Availability Statement:** The code used in this study is available at https://doi.org/10.5281/zenodo.7305224.

**Funding:** AK and JK acknowledge support by Deutsche Forschungsgemeinschaft (https://www.

## Abstract

Understanding the benefits and costs of recombination under different scenarios of evolutionary adaptation remains an open problem for theoretical and experimental research. In this study, we focus on finite populations evolving on neutral networks comprising viable and unfit genotypes. We provide a comprehensive overview of the effects of recombination by jointly considering different measures of evolvability and mutational robustness over a broad parameter range, such that many evolutionary regimes are covered. We find that several of these measures vary non-monotonically with the rates of mutation and recombination. Moreover, the presence of unfit genotypes that introduce inhomogeneities in the network of viable states qualitatively alters the effects of recombination. We conclude that conflicting trends induced by recombination can be explained by an emerging trade-off between evolvability on the one hand, and mutational robustness on the other. Finally, we discuss how different implementations of the recombination scheme in theoretical models can affect the observed dependence on recombination rate through a coupling between recombination and genetic drift.

## Author summary

Many genetic mechanisms have been invoked to explain the advantage of sex, but a coherent picture is still to emerge. Here we present a systematic theoretical and computational investigation of the effects of recombination in populations evolving on neutral fitness landscapes with unfit genotypes. We focus on populations that are large enough to be polymorphic, but nevertheless strongly affected by drift, which causes them to diffuse across the neutral network of viable genotypes. We identify a novel trade-off between evolvability, robustness and fitness that can lead to a dramatic reduction of the genetic diversity at large recombination rates. This disproves the common notion (often referred to as Weismann's hypothesis) that recombination generally increases diversity and evolvability, and instead highlights the interplay of recombination and mutational robustness.

dfg.de) through Grants No. CRC 1310 and SPP 1590. The funders had no role in study design, data collection and analysis, decision to publish, or preparation of the manuscript.

**Competing interests:** The authors have declared that no competing interests exist.

## Introduction

The neutral theory of evolution assumes that mutations have either no selective effect or are sufficiently deleterious to be quickly purged by natural selection [1]. This approximation of the distribution of fitness effects was suggested by Kimura based on observations of surprisingly high substitution rates in the amino acid sequence of certain proteins, although their function remained essentially unchanged [2, 3]. Today it is understood that the abundant neutrality in molecular evolution can arise through a wide range of mechanisms. Large portions of the genome are non-coding, allowing mutations to accumulate freely [4]. However, also in the coding regions, neutrality is prevalent due to degeneracies in the genotype–phenotype mapping at multiple levels between the blueprint, the DNA, and the final functional structure, which can be a protein, a cell, or an entire organism [5]. For example, on the scale of proteins, the degeneracies arise through synonymous mutations and through many different amino acid chains that fold to the same structure [6, 7]. On the scale of cells, neutrality is observed in regulatory gene networks [8, 9] and metabolic reaction networks [10]. Moreover, recent microbial evolution experiments [11] and theory [12] indicate that populations consistently adapt to regions of the genotype space where diminishing-returns and increasing-costs epistasis are common, which implies that the beneficial effects of mutations are almost neutral, whereas deleterious mutations typically have large negative selection coefficients. Apart from truly neutral mutations, small effect mutations can be effectively neutral if the absolute magnitude of the selection coefficient is smaller than the reciprocal of the population size [13]. Therefore neutrality is particularly important for small populations which are the subject of this study.

The assumption of a binary distribution of fitness effects, where mutations are either selectively neutral or strongly deleterious, can be conceptualized as a flat fitness landscape with holes [14, 15] or as a neutral network with varying node degrees [16, 17]. With potentially many loci at which mutations can occur, of which at least a few are selectively neutral, large clusters of viable genotypes connected by point mutations form. These clusters may span the entire sequence space and thus populations can evolve continuously without being trapped at a fitness peak [15]. In this way large neutral networks are argued to increase evolvability, since populations are able to explore large parts of genotype space leading to ever fitter genotypes [18].

However, for a complete description of evolution on neutral fitness landscapes, also the population dynamics has to be specified. A commonly used simplification is to consider the population as a point on the fitness landscape, implying that only a single genotype is present at a time. Neutral evolution then proceeds as a simple random walk on the neutral network through a sequence of fixation events [14, 19]. This scenario applies in the weak mutation regime where the mutation supply is low [20]. At the opposite end of the spectrum of evolutionary dynamics, quasispecies theory considers populations as continuously distributed clouds of genotypes in sequence space [21, 22]. An important difference compared to the weak mutation regime is that while in a simple random walk, all viable connected genotypes have the same probability of being currently occupied by the population [23], in the quasispecies regime mutationally robust genotypes, i.e. genotypes with an above-average number of viable point mutations, are preferentially occupied [16, 24]. This effect is strongly enhanced in recombining populations [25–27].

Quasispecies theory is deterministic and, strictly speaking, only applies to infinitely large populations [28]. Therefore genetic drift is absent, and all genotypes have a frequency greater than zero by definition. Moreover, in this limit, the population reaches a stationary state determined by a selection-mutation(-recombination) balance, where the frequencies of all

genotypes become constant in time. Since the number of genotypes grows exponentially with the number of loci, a shortcoming of this approximation is that it quickly becomes unrealistic for large but finite populations and is only applicable for short sequences, where the population can cover all genotypes. In this case quasispecies theory can approximate finite populations quite well [16, 25].

The purpose of this article is to describe and understand neutrally evolving finite populations in large sequence spaces for which the deterministic quasispecies limit does not apply. Within this setting, we explore a broad parameter range, such that all possible evolutionary regimes are covered. In particular, we include recombination, and study its effect across a wide range of recombination rates. We believe that such a comprehensive study, which to the best of our knowledge has not been performed previously, is essential for elucidating the conditions under which sex and recombination carry a selective advantage [29–37].

Previous work addressing the evolutionary benefits of sex and recombination has generally adopted one of two possible approaches [38]. The first approach compares sexual and asexual populations evolving in the same environment and quantifies their evolutionary performance through parameters such as the speed of adaptation or the mean fitness. Recent computational studies using this approach have investigated how epistatic interactions encoded in the fitness landscape affect the evolutionary dynamics of recombining and non-recombining populations [39–42], and experiments with microbial systems have compared the evolutionary trajectories of sexually and asexually reproducing strains [43–45]. The second approach investigates the conditions under which an allele modifying the recombination rate spreads in a population [46–48]. Here we follow the first line of research, focusing specifically on fitness landscapes with neutral and deleterious mutations.

In a recent contribution, we argued that the universal and somewhat underappreciated effect of recombination on mutational robustness may play an important role in this context [26]. Working in the deterministic limit of an infinitely large populations, we showed that mutational robustness increases monotonically with the recombination rate $r$, independent of model details, and that for a low mutation rate $\mu$ and small $r$, mutational robustness grows linearly with $r/\mu$. The deterministic limit allowed us to find precise analytical results, but many relevant questions cannot be addressed in this framework.

Here we consider finite populations in large sequence spaces. We are interested in the evolvability of the population and ask how quickly new genotypes are discovered, and how many generations it takes to discover all viable genotypes. The discovery rate of new genotypes can be crucial, e.g., if through environmental perturbations like an immune response certain genotypes become fitter over time or if the majority of the neutral network loses fitness and an *escape variant* needs to be found. While the discovery of new genotypes is therefore essential for long-term survival, the accumulation of deleterious mutations entails the risk of extinction. This induces an evolutionary trend towards increasing mutational robustness, another measure we investigate. Evolvability can also be driven by standing genetic variation within a population. Therefore we also consider different measures of genotype diversity, such as the mean Hamming distance, the number of segregating mutations and the number of distinct genotypes.

We find that certain properties like the number of segregating mutations and mean Hamming distance are independent of recombination if all genotypes are viable, but become recombination dependent when some genotypes are unfit. Other properties like the number of distinct genotypes and the discovery rate grow monotonically with $r$ if all genotypes are viable, while in the presence unfit genotypes, the dependence becomes non-monotonic. Moreover, with recombination, the discovery rate can become non-monotonic in the mutation rate, such that higher mutation rates may lead to reduced evolvability. We further discuss different

implementations of recombination in the Wright-Fisher model. Depending on the model details, recombination can act as an additional source of genetic drift which matters in small populations or large sequence spaces.

**Outline**. In the first section Models and methods, we define the structure of the genotype space and the fitness landscape. We consider both finite and infinite-sites settings. We further define the population dynamics and the implementation of recombination. Next, we introduce the relevant measures of evolvability and robustness followed by a description of the visualization of our results. In the second section Results and analysis, we first give an overview of the evolutionary regimes on neutral networks. We then explain our results in the limit of infinite sequence spaces (infinite-sites model) and continue with the results for finite sequence spaces. At the end of this section we discuss aspects of the results that show a non-robust dependence on the implementation of recombination. The results are summarized and conclusions are presented in the last section Discussion. Due to the complexity of the problem, our work relies primarily on extensive numerical simulations, but analytic results are also presented when available.

## Models and methods

### Genotype space

We consider haploid genomes with $L$ diallelic loci, which can be expressed as sequences

$$\sigma = (\sigma_1, \sigma_2, \ldots, \sigma_L) \tag{1}$$

of symbols drawn from a binary alphabet $\sigma_i = \{-1, 1\}$. This translates to a genotype space that has the properties of a hypercube $H_2^L = \{-1, 1\}^L$ of dimension $L$, where each of the $2^L$ vertices represents a genotype. Genotypes of vertices connected by an edge differ at a single locus and are therefore mutually reachable by a point mutation. The natural metric in this genotype space is the Hamming distance

$$d(\sigma^i, \sigma^j) = \sum_{k=1}^{L} (1 - \delta_{\sigma_k^i, \sigma_k^j}), \tag{2}$$

which quantifies the number of point mutations that separate two genotypes $\sigma^i$ and $\sigma^j$. For our analyses we consider both the so-called infinite-sites model (*ism*) corresponding to the limit $L \rightarrow \infty$, and the finite-sites model (*fsm*) with finite $L$. The *ism* originally introduced by [49] is easier to handle analytically, since back mutations do not occur and all mutations are novel. However, certain quantities of interest such as the mutational robustness, for which the number of viable point mutations needs to be computed, and the time until full discovery of all viable genotypes cannot be defined within the *ism*. We therefore consider both models and compare results.

### Fitness landscape

In our simulations, we either assume that all genotypes are viable, or that a fraction $1 - p$ of genotypes is unfit, such that they are quickly purged by selection. In the main text we consider viable genotypes to have fitness $w = 1$, while unfit genotypes have fitness $w = 0$, implying that they are not able to create any offspring. In this sense unfit genotypes are lethal. In S4 Fig we present results of simulations in which unfit genotypes are assigned a nonzero fitness. The results show that lethality is not a necessary condition for the effects of interest, but makes them more pronounced.

The addition of unfit genotypes strongly alters the structure of the genotype cloud and in particular the effect of recombination. To be maximally agnostic about the distribution of unfit genotypes in sequence space, we assume that each genotype is independently chosen to be viable with probability $p$ and otherwise unfit. This implies that the viable genotypes form percolation clusters on the hypercube [50], which is why we refer to the landscape as a percolation landscape. Importantly, while the fitness values assigned to different genotypes are uncorrelated in this model, on the level of alleles the percolation landscape displays strong epistatic interactions and genetic incompatibilities [15, 51].

In the case of the *fsm* we add the constraint that the network of viable genotypes is connected, i.e. that between any two viable genotypes there is a path of viable point mutations. In our simulations this is achieved by discarding all percolation landscape realizations that do not satisfy this condition; S1 Fig shows how the fraction of connected landscapes varies with $p$. The constraint is added in order to avoid situations in which the initial population is trapped in a disconnected cluster of viable genotypes. In the case of the *ism* there is no additional constraint and the fitness of a novel genotype is generated once it has been discovered by the population. A similar approach has been used previously in a study of evolvability and robustness in the absence of recombination [52].

Besides containing only minimal assumptions, we also chose the percolation landscape because its random nature makes it rich in possible structures, in the sense that it can contain regions with many viable point mutations and regions where genotypes are more often unfit and populations must evolve along a narrow fitness ridge. This makes the landscape particularly suitable to adress questions of evolvability and mutational robustness. Furthermore, the choice of this landscape has the benefit of only adding one more parameter $p$ to our analysis. From the point of view of the neutral network, the parameter $p$ determines the degree distribution.

The actual value of this parameter is expected to vary widely across levels of organismal complexity and genomic scale, but many empirical studies have identified substantial fractions of lethal or low-fitness genotypes. For example, in an empirical fitness landscape based on all combinations of 7 individually deleterious mutations in the fungus *Aspergillus niger* 27% of the genotypes were assigned zero fitness [53, 54], and similarly 40% of random mutations in the vesicular stomatitis virus were found to be lethal [55]. Correspondingly, in the present study we will explore the range $0 \leq 1 - p \leq 0.5$.

## Dynamics

To model the evolutionary forces of selection, mutation and recombination, we use individual-based Wright-Fisher models with discrete, non-overlapping generations and a constant population size $N$. For the implementation of selection and recombination, we found different computational schemes in the literature. Initial simulations showed that, whereas for large populations in the *fsm* the models become indistinguishable, the model details become apparent for small populations in the *fsm* and at arbitrary population sizes in the *ism*. In the following, the different schemes are explained. For the main part of the article, we show results for only one of the models, but mention important differences when they exist, and discuss the differences in detail in the subsection Recombination-induced genetic drift.

Fig 1 illustrates the course of one generation for three different selection-recombination schemes. In the main text, we use the model that we refer to as *concurrent recombination*. In this model, selection and recombination occur in a single step, whereas in the other two models referred to as *successive recombination* schemes these processes require two separate steps. Despite these differences, the models also share similarities, which we explain first. All models

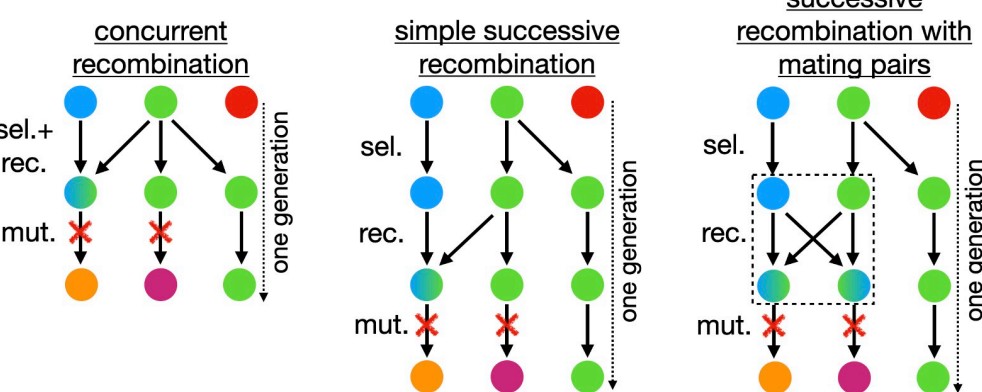

**Fig 1. One generation of evolution in the three slightly different selection-recombination-mutation schemes.**
Nodes represent individuals and colors genotypes, with green and blue nodes being viable and red nodes unfit. The
arrows show the lineages from the ancestors to the descendants. Individuals with two incoming arrows are the product
of uniform recombination. In successive recombination with mating pairs the recombining individuals are grouped in
pairs and each pair creates two descendants, which is indicated by the dashed box. Mutations are indicated by red
crosses.

have in common that an individual $j$ has two ancestors in the previous generation with a prob-
ability equal to the recombination rate $r$. In this sense the parameter $r$ can be interpreted as the
probability of sexual reproduction in a population with facultative sex [37, 56]. Unless stated
otherwise, in the event of sexual reproduction we employ a uniform crossover scheme, which
means that at each locus, the allele from one of the two ancestors $k,l$ is chosen with equal prob-
ability,

$$R: \sigma_i^j \mapsto \begin{cases} \sigma_i^k & \text{with prob. } 1/2, \\ \sigma_i^l & \text{with prob. } 1/2, \end{cases} \quad \forall\ i. \tag{3}$$

Furthermore, the mutation step always occurs last, separate from the two other processes. Dur-
ing the mutation step, each locus of each individual mutates with probability $\mu$ to the opposite
allele,

$$M: \sigma_i^j \mapsto \begin{cases} \sigma_i^j & \text{with prob. } 1-\mu, \\ -\sigma_i^j & \text{with prob. } \mu, \end{cases} \quad \forall\ i,j. \tag{4}$$

In the *ism* we take the joint limits $L \to \infty$ and $\mu \to 0$ at finite genome-wide mutation rate $U =
L\mu$. In this limit the number of mutations per individual and generation is Poisson distributed
with mean $U$. Importantly, each mutation is then novel and back mutations cannot occur.
This can be expressed by characterising each individual's genotype by the set of acquired novel
mutations, e.g.

$$M: \sigma^j = \{\tau, \zeta\} \to \begin{cases} \{\tau, \zeta\} & \text{with prob. } e^{-U}, \\ \{\tau, \zeta, \lambda\} & \text{with prob. } Ue^{-U}, \quad \forall\ j, \\ \ldots \end{cases} \tag{5}$$

denoted by Greek letters. It is necessary to track the mutations carried by each invididual also
in the *ism* in order to be able to implement recombination, which combines mutations and

breaks them apart. However, once a mutation is fixed in the population, i.e., it is present in all individuals, it can be omitted due to the lack of back mutations, thereby keeping the list of stored mutation finite. Also, the stored list of fitness values of discovered genotypes can be purged of those genotypes whose mutation set does not contain newly fixed mutations, as they cannot be reached anymore.

While the features discussed so far are the same in all models, the differences are the following.

**Concurrent recombination.** In this case all individuals that are not the product of a recombination event select one ancestor with a probability proportional to the ancestor's fitness. Simultaneously those individuals that are the product of a recombination event select two different ancestors with a probability proportional to their fitness.

**Successive recombination.** In both successive recombination models, selection occurs first, where all individuals choose one ancestor according to their fitness. Next, recombination takes place independent of fitness for which two implementations are possible:

- In the simple successive recombination model, individuals that are a product of recombination simply choose two different ancestors that survived selection.

- In the successive recombination with mating pairs model, all individuals that survived selection and happen to recombine are pooled in groups of two, which then create two offspring individuals. These two offspring individuals are complementary in their recombined material, that is, if one offspring has the allele of the first parent, the other offspring will have the allele of the second parent at the corresponding locus.

Importantly, the three recombination schemes differ in the way in which recombination couples to genetic drift. In the concurrent recombination and the successive recombination with mating pairs model, genetic drift is independent of the recombination rate $r$, but this is not the case for the simple successive recombination model (see S1 Appendix). Recombination-dependent drift is a confounding factor that needs to be taken into account in the interpretation of the results of the latter model. Since the concurrent and the successive recombination with mating pairs models are implemented in two commonly used open software packages [57, 58], we stick to a non-recombination dependent genetic drift model in the main text and only occasionally refer to differences that would otherwise appear. The recombination-dependent genetic drift model has been used by [40]. Being aware about these differences might be important for the design of experiments, e.g., in the context of *in vitro* recombination [59]. Of the two non-recombination dependent genetic drift models, we choose the concurrent recombination model because it is somewhat simpler. In particular, in this model the number of recombining individuals does not have to be an even number.

## Measures of evolvability and robustness

To quantify evolvability in the *ism* we consider the discovery rate $r_{dis}$ of novel genotypes and the fixation rate $r_{fix}$ of mutations. The discovery rate $r_{dis}$ is the average number of novel viable genotypes that are discovered in each generation, either through mutation or recombination. The fixation rate $r_{fix}$ of mutations measures the average number of segregating mutations that become fixed in each generation.

In the *fsm* we monitor evolvability through the time $t_{fdis}$ and the number of mutation events $N_{mut}$ until full discovery. Starting from a monomorphic population carrying a single randomly selected viable genotype, we say that full discovery is reached when all viable genotypes have been present in at least one individual in at least one generation. The time is measured in

generations and a mutation event occurs if an individual acquires one or multiple mutations during reproduction.

Evolvability in terms of standing genetic diversity is characterized through several well-known measures of population genetics, for which the time average is descriptive for the randomly drifting genotype cloud (note that in the regimes of interest here, the population cannot attain an equilibrium state, because the genotype space is larger than the population). Such measures are the pairwise mean Hamming distance between two individuals in the population

$$d_{pw} = \frac{1}{N(N-1)} \sum_{\substack{i,j \\ i \neq j}} d(\sigma^i, \sigma^j),$$

(6)

the number of viable distinct genotypes

$$Y = |\{\sigma^i | i \in 1, 2, \ldots, N \wedge \sigma^i \text{is viable}\}|,$$

(7)

and the number of segregating mutations

$$S = \sum_{i}^{L} \left( |\{\sigma_i^j\}_{j \in 1,2,..,N}| - 1 \right),$$

(8)

i.e., the number of loci at which both alleles are present in the population. These measures are used for the *ism* as well as for the *fsm*.

Additionally, for the *fsm* we consider the mutational robustness $m$ of the population. The robustness $m_{\sigma^i}$ of an individual $i$ is equal to the fraction of viable point mutations of its genotype $\sigma^i$ if it is itself viable, and equal to 0 otherwise. The mutational robustness of the population is the average robustness of all individuals,

$$m = \frac{1}{N} \sum_{i}^{N} m_{\sigma^i}.$$

(9)

This quantity depends on the population distribution in genotype space, and increases if the population moves to genotypes for which most point mutations are viable. Apart from these measures for evolvability and mutational robustness we also measure the mean fitness defined by

$$\overline{w} = \frac{1}{N} \sum_{i}^{N} w_{\sigma^i}.$$

(10)

For the assumed binary fitness distribution, the mean fitness is determined by the mutation and recombination load, i.e. the fraction of mutation and recombination events per generation that give rise to unfit genotypes with fitness $w = 0$. Therefore we also measure the average viable recombination fraction, which represents the fraction of recombination events per generation that generate a viable genotype. This can be considered as a measure of recombination robustness which complements the measure of mutational robustness.

Except for the measures for evolvability in the *fsm*, our numerical results always show the averages in steady state. For the *ism*, we evolve and evaluate the population for $t = \sqrt{10^{10}/U}$ generations for each data point. For the *fsm*, we evolve the population until all genotypes have been discovered once and, except for the measures of evolvability, more than $10^5$ mutation events have occurred. This is done for $10^4$ landscape realizations and for each data point. We measure all quantities at the end of a generation, i.e., after the mutation step, and denote the averages of measured quantities by overbars.

## Illustration of results

**3D wireframe plots.**   In order to represent the numerical results comprehensively, we mostly use 3D wireframe plots. In these plots, the wireframe lines run along either constant recombination rate or constant mutation rate to guide the eye. The color of the wireframe lines depends on their height. Additionally a contour plot is shown below the wireframe. The viewing angles vary and are selected such that the results can be seen in the best possible way.

**Graph representation.**   To visualize the population distribution in sequence space, we use a graph representation. In this graph each genotype in the population is represented as a node, and nodes whose genotypes differ by a single point mutation are connected by an edge. Therefore, the resulting graph only contains information about nearest neighbor relationships. However, as long as the genotype cloud is not distributed too broadly in sequence space, we expect most nodes to have at least one edge, thereby forming clusters of connected components in the high dimensional sequence space. To arrange the nodes in two dimensions we use a forced-based algorithm called *ForceAtlas2* [60]. This leads to a configuration in which nodes that share many edges form visual clusters. The frequencies of the genotypes are represented by the node sizes.

## Results and analysis

### Evolutionary regimes

To organize the discussion of the results, we recall here the distinct evolutionary regimes that can be realized on neutral networks. In the *monomorphic* or *weak mutation* regime $NU \ll 1$, mutations are rare and either fix with probability $1/N$ or go extinct through genetic drift before another mutation originates. In this regime, the population consists of a single genotype most of the time and can be described by a "blind ant" random walker [16, 23]. Without lethal genotypes, a step to one of the current genotype's mutational neighbors is taken with equal probability at rate $U$ independent of $N$. With lethal genotypes, a step is discarded when the randomly chosen point mutation is lethal. On a connected network this implies that the population occupies on average each genotype, irrespective of its degree, with equal probability [23]. In this regime, the effect of recombination is minimal since recombination is fueled by combining segregating mutations, which do not exist most of the time. Furthermore, as long as there are not more than two genotypes at Hamming distance 1 from each other in the population, the population is by definition in linkage equilibrium. Nonetheless, in subsection Recombination-induced genetic drift we show that recombination can have an effect in this regime if it couples to the genetic drift.

In the *polymorphic* regime ($NU \geq 1$), where the population is a cloud of competing genotypes, two subregimes can be distinguished. If the population size is large compared to the number of genotypes ($N \gg 2^L$), all viable genotypes can become occupied and an equilibrium state is reached. In this case genetic drift is irrelevant and the equilibrium distribution can be described by assuming deterministic dynamics of quasispecies type. In the absence of unfit genotypes, all genotypes then have the same frequency in the population, similar to the monomorphic regime. Importantly, though, with unfit genotypes, the population distribution becomes non-uniform, in that robust viable genotypes that have less deleterious point mutations exhibit higher frequency [16]. This imbalance increases dramatically with increasing recombination rate [26].

Of particular interest in the context of the present work is the second polymorphic subregime, where the population size is smaller than the number of genotypes ($N \ll 2^L$) or even smaller than the number of loci ($N \ll L$), and the population clearly cannot attain an

equilibrium state. Instead, it will diffuse as a cloud of genotypes on the neutral network. This subregime has, to the best of our knowledge, not been fully covered in the literature, in particular in the presence of recombination. In the following sections, we first study the population structure in the *ism*, where $N \ll L$ is guaranteed by definition. Subsequently we consider a finite number of sites and study the *fsm* for $N \ll 2^L$.

## Infinite-sites model

In this section, we present numerical results that show the impact of recombination and lethal genotypes on the population structure in the *ism*. We keep the population size fixed at $N = 100$ and discuss the dependence on the recombination rate $r$, the mutation rate $U$ and the fraction of viable genotypes $p$.

**Discovery rate.**  Fig 2 displays the discovery rate of formerly unexplored viable genotypes for $p = 1$ and $p = 0.5$. Without recombination, the discovery rate is given by

$$r_{dis} = pN(1 - e^{-U}) \overset{U \ll 1}{\approx} pNU \tag{11}$$

since each mutation generates a new genotype which is viable with probability $p$. The numerical results show that the effect of recombination on the discovery rate depends on the mutation rate $U$, the product $NU$ and the fraction of viable genotypes $p$. If $NU \ll 1$, the effect of recombination is minimal, since there are very few segregating mutations that can be recombined. However, for $NU \geq 1$ we notice a rather complex dependence. In the absence of lethal genotypes ($p = 1$), recombination increases the discovery rate monotonically. The relative increase is maximal if $U \ll 1$ but $NU \gg 1$. If $U$ is of order 1, the effect of recombination becomes smaller since the maximum discovery rate is capped by the population size $N$ and almost exhausted through mutations.

In the presence of lethals ($p = 0.5$) the behavior is more surprising. For small $U$, recombination has almost no effect, even if $NU \geq 1$. As $U$ increases, an intermediate recombination rate becomes optimal for the discovery rate. This intermediate peak shifts to larger $r$ with increasing $U$ and the drop-off becomes sharper until $U$ is of order 1, where the intermediate peak

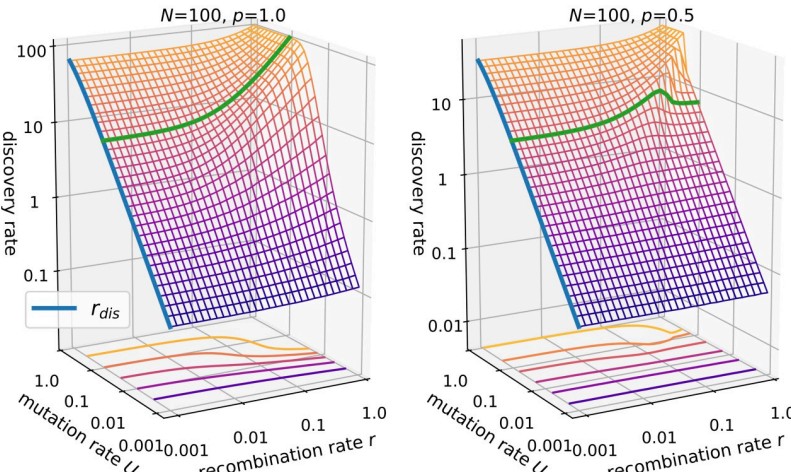

**Fig 2. Discovery rate in the *ism*.** The population size is $N = 100$ with either $p = 1.0$ (left panel) vs. $p = 0.5$ (right panel). The green line at $U = 0.1$ highlights the different effects of recombination in the absence ($p = 1$) and presence ($p = 0.5$) of lethal genotypes. The blue lines in both panels show Eq 11.

vanishes and the behavior is again similar to $p = 1$. These results indicate three regimes for the effect of recombination in the *ism*: (i) $NU \ll 1$, (ii) $NU \geq 1$ and $U \ll 1$, and (iii) $U \approx 1$.

**Fixation rate.**   Next, we consider the fixation rate of segregating mutations. In the absence of recombination mutations are expected to fix at rate

$$r_{fix} = \frac{pU}{e^{-U} + p(1 - e^{-U})} \overset{U \ll 1}{\approx} pU.$$

(12)

To arrive at this expression, we first note that in each generation $pNU$ novel viable mutations are acquired by individuals of the population. However, only one individual becomes the ancestor of the whole population, which implies that only a fraction of the novel mutations reach fixation. The probability that one individual becomes the ancestor of the whole population is equal to the inverse of the average number of viable individuals each generation, which is given by $Ne^{-U} + Np(1 - e^{-U})$.

The results in Fig 3 confirm this expectation in the absence of recombination. Moreover, results show that concurrent recombination has no effect if all genotypes are viable. However, for $p = 0.5$ the fixation rate is seen to dramatically decline at large recombination rates when $NU \geq 1$ and $U \ll 1$. This disruption of fixation is released only when the mutation rate becomes of order 1.

**Number of distinct genotypes.**   The number of distinct genotypes $Y$ is naturally closely related to the discovery rate, since with more novelty discovered in each generation, more distinct genotypes should accumulate, cf. Fig 4. An analytical expression for the case of no recombination and no lethal genotypes was derived in [61]:

$$\overline{Y}(r = 0) = \sum_{i=0}^{N-1} \frac{\theta}{\theta + i} \quad \text{with} \quad \theta = 2NU.$$

(13)

Our numerical results fit this expression and furthermore show that in the absence of recombination, the formula can be extended to include lethal genotypes by replacing $\theta$ with

$$\theta^* = 2pNU.$$

(14)

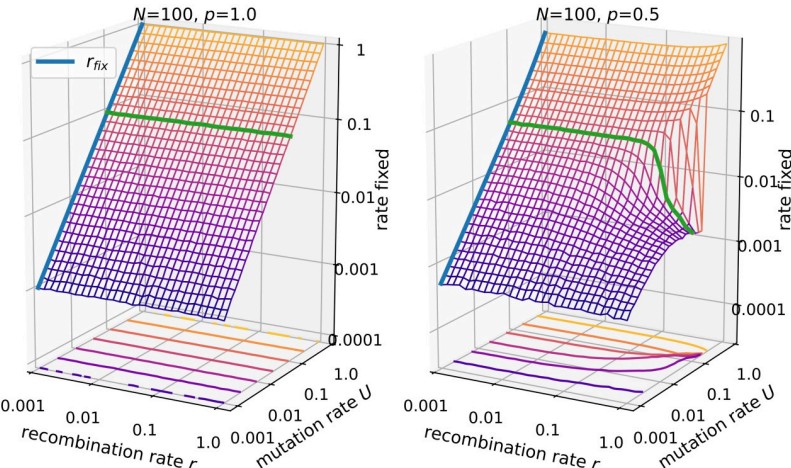

**Fig 3. Fixation rate of segregating mutations in the *ism*.** Parameters are $N = 100$ and $p = 1.0$ (left panel) vs. $p = 0.5$ (right panel). The green line is drawn at $U = 0.1$. The blue line represents Eq 12.

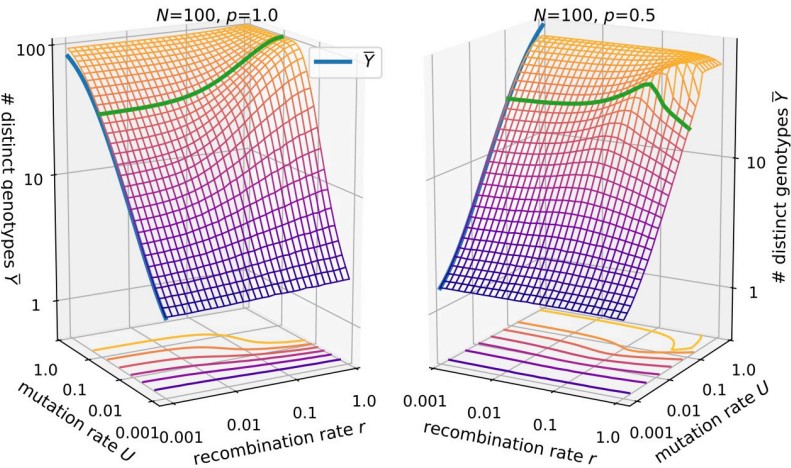

**Fig 4. Number of distinct genotypes in the *ism*.** Parameters are $N = 100$ and $p = 1.0$ (left panel) vs. $p = 0.5$ (right panel). The blue lines shows Eq 13, where $\theta$ is replaced by $\theta^*$ for $p \leq 1$. The green line is drawn at $U = 0.1$.

The reasoning behind this replacement is that $\theta$ is related to the rate of arrival of new neutral mutations each generation, with the factor 2 being model dependent. In general, the effect of recombination on the number of distinct genotypes is similar to that on the discovery rate.

**Number of segregating mutations.** We also consider the number of segregating mutations $S$ (Fig 5). These span an effective sequence space of size $2^S$ in which recombination can create novel genotypes. An analytical expression for the number of segregating mutations in the *ism* was developed by [62], again assuming no recombination and no lethal genotypes. It can be derived from the expectation of the total length of the genealogical tree to the most recent common ancestor, which is given by [63]

$$\overline{T}_{total} = 2N \sum_{i=1}^{N-1} \frac{1}{i} \tag{15}$$

for the Wright-Fisher model. The total tree length is equal to the total time, and multiplying by

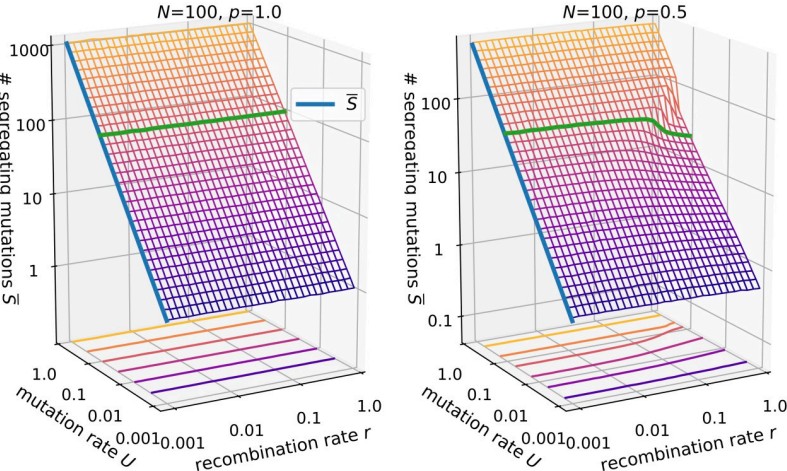

**Fig 5. Number of segregating mutations in the *ism*.** Parameters are $N = 100$ and $p = 1.0$ (left panel) vs. $p = 0.5$ (right panel). The blue lines shows Eq 16. The green line is drawn at $U = 0.1$.

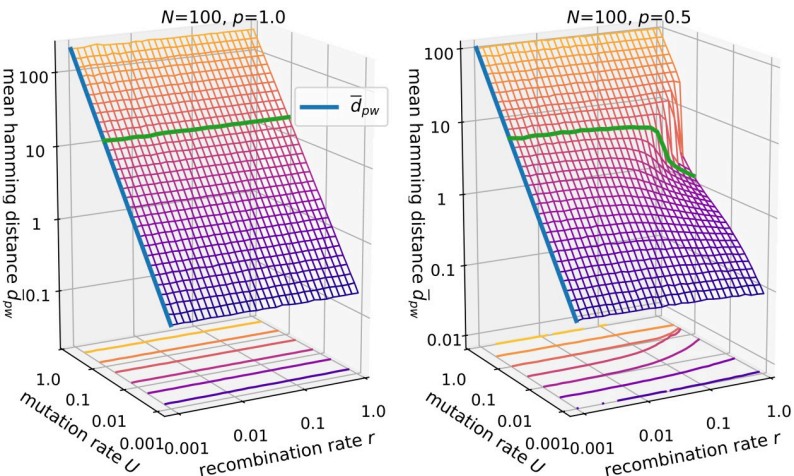

**Fig 6. Mean Hamming distance in the *ism*.** Parameters are $N = 100$ and $p = 1.0$ (left panel) vs. $p = 0.5$ (right panel). The blue lines shows Eq 17. The green line is drawn at $U = 0.1$.

the mutation rate $U$ and the fraction of viable genotypes yields the average number of segregating mutations

$$\overline{S} = \theta^* \sum_{i=1}^{N-1} \frac{1}{i}. \tag{16}$$

The results also show that for $p = 1$ and the concurrent recombination model used here, the number of segregating mutations is independent of $r$. This is not generally true but depends on the implementation of recombination (see Recombination-induced genetic drift). For $p = 0.5$ and $NU \geq 1$, $U \ll 1$, recombination generally decreases the number of segregating mutations. For example, at $U = 0.1$ the number decreases from around 50 to about 20, which still yields an enormous effective sequence space compared to the population size. This alone cannot explain the decrease in evolvability seen in Figs 2 and 4.

**Mean Hamming distance.**   Since recombination occurs between pairs of individuals we now consider the pairwise mean Hamming distance $\overline{d}_{pw}$ (Fig 6). From Eq 15 we conclude that the mean length of the genealogical tree to the most recent common ancestor for two random individuals is given by $2N$, which directly leads to the expression

$$\overline{d}_{pw} = \theta^* \tag{17}$$

for the mean Hamming distance in non-recombining populations. The results for $p = 0.5$ show that recombination contracts the population cloud in sequence space in the presence of lethal genotypes. The functional relationship is similar to the number of segregating mutations but the relative change is more dramatic, e.g., for $U = 0.1$ the distance drops from 10 to about 1.

**Cross section of the population cloud.**   To further understand the contraction in sequence space, we took a cross section of the population cloud by measuring the Hamming distance of each individual to the ancestral genotype that contains only fixed mutations, averaged over many generations (Fig 7). This quantity is equal to the number of segregating mutations in an individual. For $r = 0$ the Hamming distance distribution is seen to follow a hypoexponential distribution, which converts to a Poisson distribution for large $r$. The hypoexponential distribution follows from the correspondence between the Hamming distance and

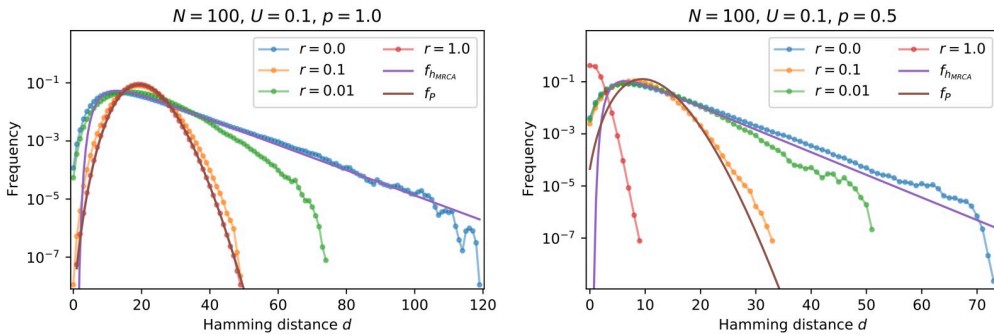

**Fig 7. Cross section of the population cloud in the *ism*.** Parameters are $N = 100$, $U = 0.1$ and $p = 1.0$ (left panel) vs. $p = 0.5$ (right panel). The figures show the distribution of the Hamming distance to the genotype containing only fixed mutations. The hypoexponential distribution $f_{h_{MRCA}}$ is given by Eq 18 and $f_P$ is a Poisson distributions with mean $\theta^*$. The data were accumulated over $10^6$ generations.

the time to the most-recent-common-ancestor $T_{MRCA}$, the distribution of which is well known [63]. With a mutation rate $U$ and a fraction $p$ of viable genotypes, this yields the distribution

$$f_{h_{MRCA}}(d) = \frac{2}{\theta^*} \sum_{i=2}^{N} \binom{i}{2} e^{-\binom{i}{2}\frac{2d}{\theta^*}} \prod_{j=2, j\neq i}^{N} \frac{\binom{j}{2}}{\binom{j}{2} - \binom{i}{2}}, \tag{18}$$

with mean $\theta^*$. At high recombination rates, segregating mutations become well mixed among all individuals, and the number of segregating mutations acquired by an individual follows a Poisson distribution. For $p = 1$ the mean is independent of $r$ and equal to $\theta^*$, but for $p = 0.5$ we observe a strong contraction of the distance distribution at $r = 1$. This shows that the focal genotype, around which the contraction occurs, is the ancestral genotype that contains only fixed mutations.

**Mean fitness and recombination load.** The contraction of the genotype distribution described in the preceding paragraphs can be interpreted in terms of a reduction of the recombination load. The rate of lethal mutations is always $U(1 - p)$ and cannot be optimized in the *ism*. Contrary to that, the outcome of recombination events depends on the genotype composition of the population. If the population is contracted around a focal genotype and most individuals are closely related to each other, the effective sequence space for recombination is much smaller than $2^S$. Therefore it becomes likely that a recombination event will not create a novel genotype but a genotype that already exists in the current genotype cloud and that, more importantly, is viable, which increases the mean fitness. In other words, large recombination rates lead to selection against rare genotypes that are distant to the focal genotype, because they are more likely to produce lethal genotypes if they recombine with the focal genotype. This is ultimately a consequence of the genetic incompatibilities in the percolation landscape [15, 51]. The correlated response of recombination load and mean fitness to an increase in recombination rate is shown in Fig 8. If the population is sparsely distributed which happens at high mutation rates, most recombination events create novel genotypes, which leads to a viable recombination fraction equal to $p$. Contrary to that in a monomorphic population ($NU \ll 1$) no novelty results from recombination. In the regime $NU \geq 1$, $U \ll 1$, e.g. at $U = 0.1$, we see that with increasing $r$ the viable fraction initially decreases, as the population becomes more diverse, leading to a decrease in fitness. At large recombination rates this trend reverses as the population becomes concentrated around a focal genotype, which then also leads to a fitness increase.

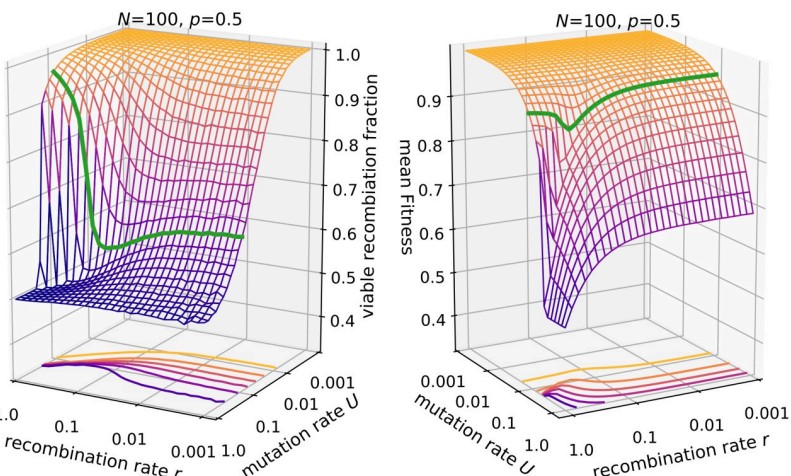

**Fig 8. Viable recombination fraction and mean fitness in the *ism*.** Parameters are $N = 100$, $p = 0.5$. The green line is drawn at $U = 0.1$.

**Fraction of unfit genotypes.**   The results presented so far were obtained for the two values $p = 1$ and $p = 0.5$ of the fraction of viable genotypes. Fig 9 shows cross sections of the previously shown 3D plots at either fixed recombination rate $r = 1$ (left column) or fixed mutation rate $U = 0.1$ (right column) and four different values of $p$. For fixed $r = 1$, the lethal genotypes strongly reduce evolvability by contracting the genotype cloud when the mutation rate is low, but the contraction is released once the mutation rate is strong enough. At this point the population evolves independent of fitness and therefore the measures coincide with the results for $p = 1$. However, the mean fitness is strongly reduced and equal to $p$. With smaller $p$ this transition happens at larger $U$ and strikingly at small enough $p$ the numerical results display a discontinuity as a function of $U$. The dependence on mutation rate resembles the error threshold phenomenon of quasispecies theory, in which the population delocalizes from a fitness peak in a finite-dimensional sequence space when the mutation rate is increased above a critical value [21, 22]. In the quasispecies context it has been shown that error thresholds do not occur in neutral landscapes with lethal genotypes, at least not in the absence of recombination [28, 64]. Moreover, for non-recombining populations the mean population fitness is generally continuous at the error threshold, whereas recombination can induce discontinuous fitness changes and bistability [65]. Although the transfer of results from the infinite population quasispecies model with finite sequence length $L$ to the finite population *ism* is not straightforward, our results are generally consistent with previous work in that there is no discontinuity in the absence of recombination, while at sufficiently large recombination rates we find evidence for a discontinuous error threshold in the percolation landscape with lethal genotypes. From the perspective of quasispecies theory, the shift of the transition to larger $U$ for decreasing $p$ may tentatively be explained as an effect of increased selection pressure in landscapes with a larger fraction of lethal genotypes.

The results for fixed mutation rate $U = 0.1$ displayed in the right column of Fig 9 show that the contraction of the population only occurs if the recombination rate is sufficiently large, whereas otherwise recombination increases evolvability. Importantly, with an increasing fraction of lethal genotypes, the contraction occurs at lower recombination rates. S2 Fig demonstrates that for smaller mutation rates $U$, smaller values of $r$ are sufficient to contract the population. This fits the expectation based on the analysis of [26], where it was demonstrated

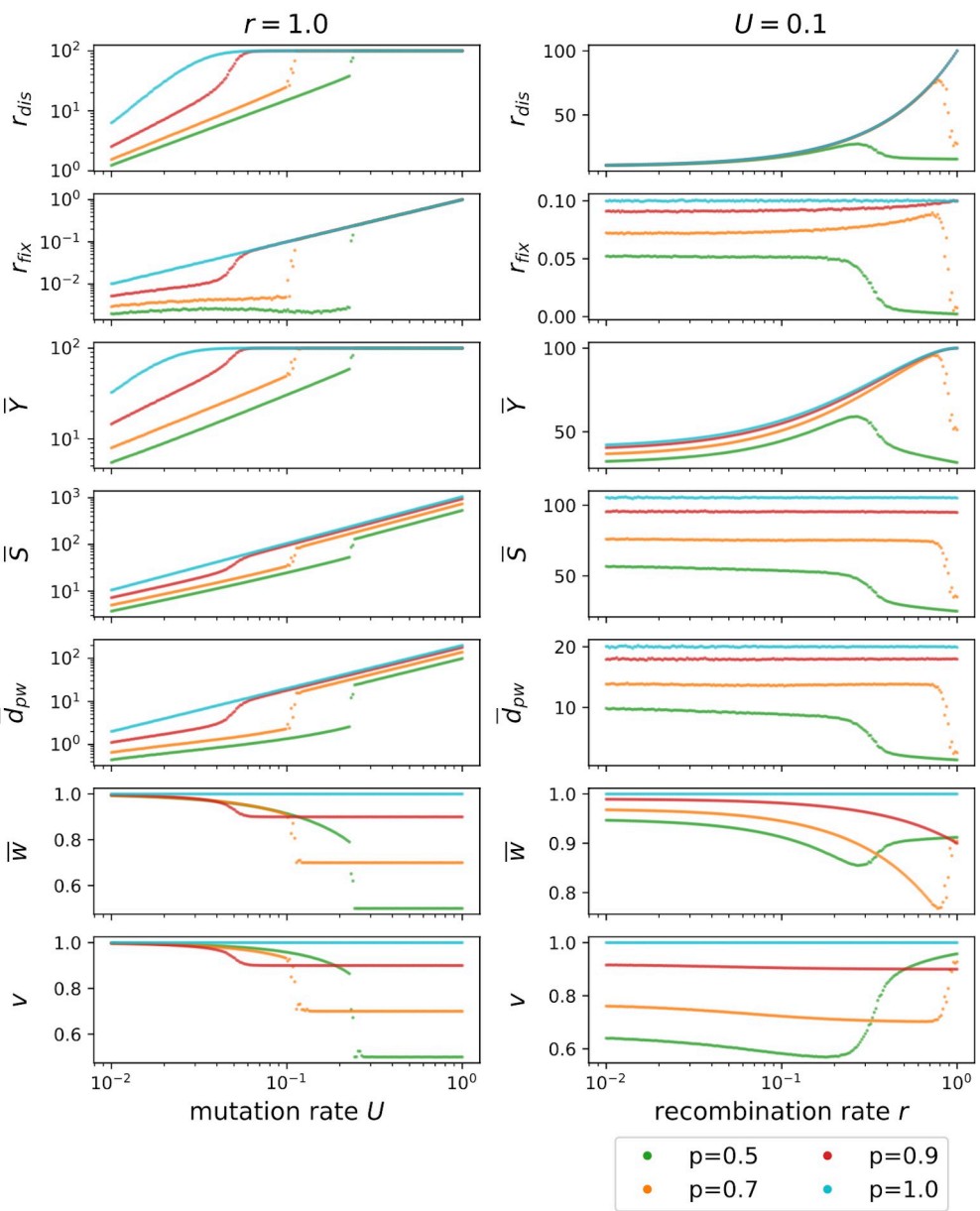

**Fig 9. Various measures characterizing the *ism* for four different values of the fraction *p* of viable genotypes.** The left column shows the dependence on the mutation rate *U* for fixed *r* = 1.0, and the right column shows the dependence on the recombination rate *r* at fixed *U* = 0.1. The last row shows the fraction *v* of viable genotypes created by recombination events. The population size is *N* = 100. Two supplementary figures display results corresponding to the right column of the figure at lower mutation rate *U* = 0.05 (S2 Fig) and with varying fitness of the unfit genotypes (S4 Fig).

that the scale for the effect of the recombination rate on mutational robustness is set by the mutation rate.

**Large populations.**    For large population sizes the number of segregating mutations grows rapidly, such that storing the part of the fitness landscape that could be revisited through recombination becomes computationally challenging. This limits the range of population sizes that can be explored. Nevertheless, the results for the discovery rate for *N* = 1000 displayed in

S3 Fig suggest that, for large populations, the interesting regime with a non-monotonic behavior in $r$ appears in an even larger range of mutation rates than expected from the conditions $U \ll 1$ and $NU \geq 1$.

**Robustness of the results.** In order to investigate the robustness of the results for the *ism*, we relaxed several conditions. S4 Fig illustrates that unfit genotypes need not be strictly lethal for the effects discussed to occur. Moreover, we investigated variations of the recombination mechanism. On the one hand, in S5 Fig we employed a one-point crossover scheme instead of the uniform crossover. On the other hand, in S6 Fig we considered populations with obligate sexual reproduction, where the recombination rate $r$ defines the average number of crossovers instead of the average fraction of individuals that recombine. For both cases, the results show that the non-monotonic behavior persists.

**Summary of *ism* results.** As expected, for $NU \ll 1$ or $U \approx 1$, recombination has almost no effect since the population is either monomorphic or dominated by mutations. In contrast, for $NU \geq 1$ and $U \ll 1$ the behavior is rather complex. While low recombination rates generally diversify the population and increase the discovery rate, the population can dramatically change its genotype composition at high recombination rates, such that most genotypes are tightly clustered around a focal genotype. In the percolation landscape, the onset of this structural change depends on the fraction of lethal genotypes, whereas for general fitness landscapes we expect it to be determined by the degree distribution of the neutral network. The focal genotype of a tightly clustered population contains no segregating mutations. As a consequence the clustering decreases the discovery rates as well as the fixation rate, but the mean fitness increases. We conclude that recombination does not generally lead to increased evolvability, but may instead reduce diversity in order to regain mean fitness.

## Finite-sites model

In this section, we study the effect of recombination in the finite-sites model. In contrast to the *ism* back mutations are possible, and the number of viable point mutations varies between genotypes. Therefore the population can optimize its mutational robustness to increase fitness. We keep the population size and sequence length fixed at $N = 100$ and $L = 10$, respectively, and investigate the interplay between the mutation rate per site $\mu$ and the recombination rate $r$.

**Mutational robustness.** The results for the mutational robustness displayed in Fig 10 show a complex dependence on $r$ and $\mu$, which reflects the different evolutionary regimes described above. Similar to the *ism*, for $NL\mu \ll 1$ the population is essentially monomorphic. In this regime it behaves like a random walker and all genotypes have equal occupation probability, such that the mutational robustness is equal to the average network degree $p = 0.5$. For a monomorphic population, recombination has no effect. With increasing mutation rate ($NL\mu \geq 1, \mu \ll 0.5$), the population becomes polymorphic but also more mutationally robust. Strikingly, with recombination, this effect is strongly amplified, as was observed previously in the quasispecies regime [25, 26]. Similar to the results presented for the *ism* (Fig 8), the increase in mutational robustness is accompanied by an increase in mean fitness (S7 Fig). In [26] we showed that mutationally robust genotypes are more likely to be the outcome of recombination events, because they have a larger share of potentially viable parents. Therefore increased mutational robustness is a universal feature of recombining populations.

However, even higher mutation rates ($\mu \approx 0.5$) are detrimental to robustness, and recombination then also has a slightly negative effect. In this regime, the population is not concentrated anymore on a focal genotype but becomes highly delocalized and almost independent of the previous generation. Because of this, recombination events will produce random genotypes; note that for $\mu = 0.5$, all genotypes have the same probability after mutation, independent of

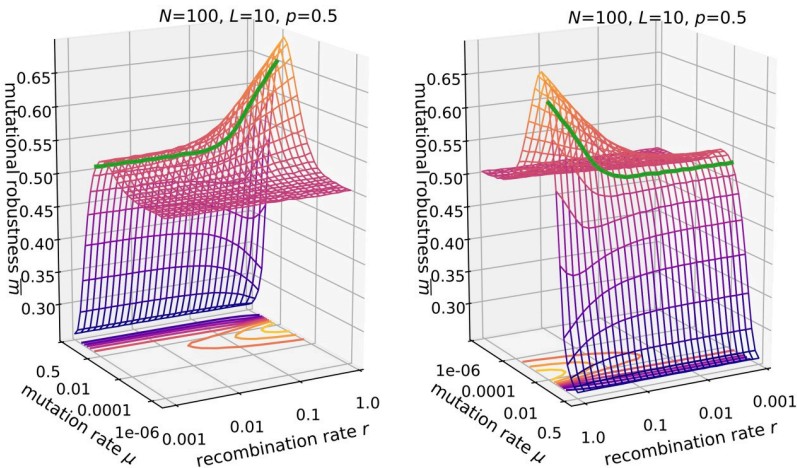

**Fig 10. Mutational robustness in the *fsm*.** Parameters are $N = 100$, $L = 10$, $p = 0.5$. The two panels show the same data from two different viewing angles. The green line is drawn at $L\mu = 0.1$.

viability. Thus, similar to the *ism* we can define three regimes with qualitatively different effects of recombination: (i) $NL\mu \ll 1$, (ii) $NL\mu \geq 1$ and $\mu \ll 0.5$, and (iii) $\mu \approx 0.5$.

**Time to full discovery.** We quantify the discovery rate in the *fsm* in terms of the time until all genotypes have been discovered, $t_{fdis}$. For ease of comparison with the results for the *ism* illustrated in Figs 2 and 9, we consider in the following the reciprocal $1/t_{fdis}$ which can be interpreted as a discovery rate. Comparison of the *fsm* results in Figs 11 to 2 shows that the overall behavior is similar, but that a potential benefit of recombination in terms of increased evolvability is very much reduced for $p = 1$ and $p = 0.5$ in the *fsm*. Instead the decrease in the discovery rate for large $r$ at $p = 0.5$ is much more pronounced than in the *ism*. Whereas in the *ism* the discovery rate never drops below its value in the absence of recombination ($r = 0$), here the time to full discovery *diverges* at large recombination rates when $NL\mu \geq 0$ and $\mu \ll 1$. Furthermore, the dependence on mutation rate becomes non-monotonic at large $r$, which does not happen in the *ism*. The increase in the time to full discovery coincides with increased

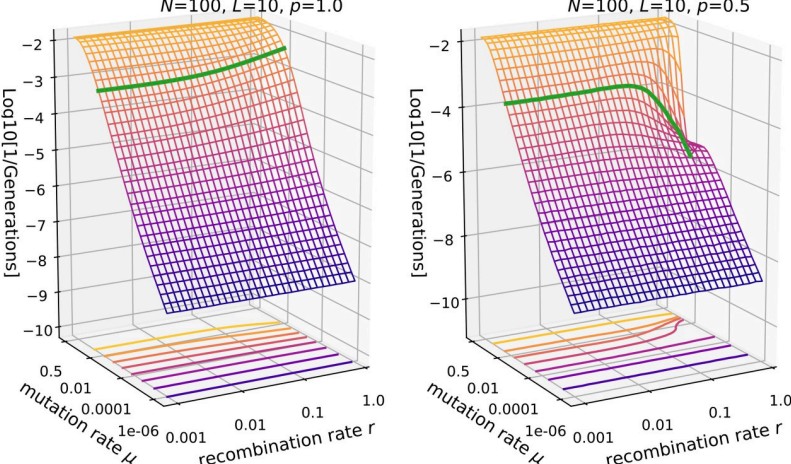

**Fig 11. Reciprocal of the time to full discovery in the *fsm*.** Parameters are $N = 100$, $L = 10$ and $p = 1$ (left panel) vs. $p = 0.5$ (right panel). The green line is drawn at $L\mu = 0.1$.

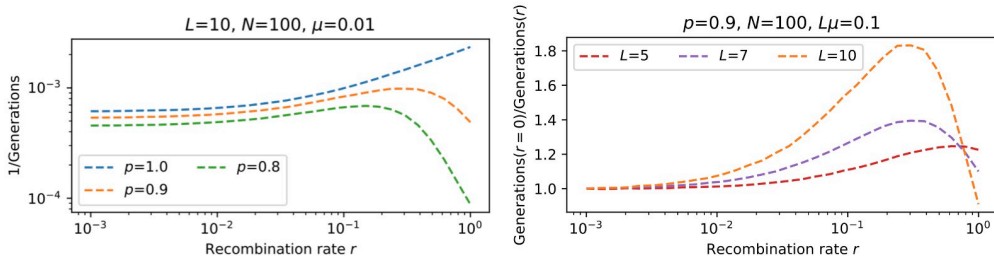

**Fig 12. Fraction $p$ and sequence length $L$ impact on the time to full discovery in the *fsm*. A**: Reciprocal of the time to full discovery is shown as a function of recombination rate for different values of the fraction $p$ of viable genotypes. **B**: Relative change in the reciprocal of the time to full discovery for different values of the sequence length $L$ and fixed genome-wide mutation rate $U = L\mu$.

mutational robustness (Fig 10) and a large viable recombination fraction (S7 Fig). Therefore this divergence occurs because the population is focused and entrenched in the highly robust regions of the fitness landscape. In this way a fitness ridge surrounded by many lethal genotypes can become almost impassable for strongly recombining populations.

In Fig 12 this phenomenon is explored over a wider range of parameters. When the fraction of viable genotypes is closer to $p = 1$, the rate of discovery displays an intermediate maximum as a function of $r$ (panel A), and this behavior becomes more pronounced for longer sequences (panel B). This demonstrates, that in terms of the time to full discovery already a small fraction of unfit genotypes $1 - p$ is sufficient to create a non-monotonic dependence on $r$.

In S8 Fig we further investigated a scenario in which a certain *escape variant* needs to be found by at least one individual. As before the initial genotype is chosen randomly among all viable genotypes. The results show that the overall behavior is similar to the time until full discovery. In this setting we could further investigate how the discovery time depends on the robustness of the randomly chosen escape variant and the initial genotype. The figures illustrate that the robustness of the initial genotype is irrelevant, but more robust escape variants are found more quickly. This observation does not change with recombination but is more pronounced at lower recombination rates.

**Number of mutation events until full discovery.** As another measure of evolvability we consider the total number of mutation events $N_{mut}$ until all viable genotypes have been discovered (S9 Fig). In the random walk regime $NL\mu \ll 1$, $N_{mut}$ is independent of $\mu$ and $r$, since each mutation has the probability $1/N$ to go to fixation. As the mutation rate increases and the population spreads over the genotype space, fewer mutation events are necessary for full discovery. Similar to the time to full discovery $t_{fdis}$, depending on the fraction of viable genotypes, recombination can be either beneficial or detrimental for evolvability. In fact the two measures are related by

$$N_{mut} = t_{fdis}NL\mu \tag{19}$$

as long double mutations, which we count as a single mutation event, are sufficiently rare ($L\mu \ll 1$).

**Genetic diversity.** S10 Fig summarizes results for the genetic diversity in the *fsm*. Overall the impact of recombination on the genetic diversity is similar to, but less pronounced than the results for the *ism*. In particular, no discontinuities are observed in the variation of diversity measures with mutation rate (compare to Fig 9). Because the number of segregating mutations and the mean Hamming distance are bounded in the *fsm*, the capacity of recombination for creating diversity is limited. For example, while in the *ism* there are around 100 segregating

mutations for $U = 0.1$, in the *fsm* with $L\mu = 0.1$ this number is limited by the sequence length $L = 10$ (middle row of S2 Fig). This suggests that the non-monotonic effect of recombination on evolvability in the *fsm* for $p \leq 1$ is mostly caused by an increase in mutational robustness. For $p = 1$ an analytical expression for the mean Hamming distance is derived in the S1 Appendix. For the concurrent recombination scheme the result

$$\overline{d}_{pw} = \frac{2(1 - \mu)\mu L N}{4(1 - \mu)\mu(N - 1) + 1} \tag{20}$$

is independent of the recombination rate, but this property is model dependent (see Recombination-induced genetic drift for further discussion).

**Graph representation.**    To further illustrate the genotype composition of the population, a graph representation is employed in Fig 13. These graphs show snapshots of genotype clouds that have evolved for sufficiently many generations to be independent of the initial condition. The population parameters are chosen to be in the regime $NL\mu \geq 1$, $\mu \ll 0.5$. The graphs of the recombining populations consist of significantly more edges representing mutational neighbors, which implies that they form a densely connected component in sequence space. By comparison, the graphs of the non-recombining populations have fewer edges, which implies that the populations are more dispersed. This is consistent with the results for the cross section of the genotype cloud in the *ism* in Fig 7, which show a narrower distribution for recombining populations. S11 Fig shows genotype clouds for larger values of $L$ and $N$, but within the same

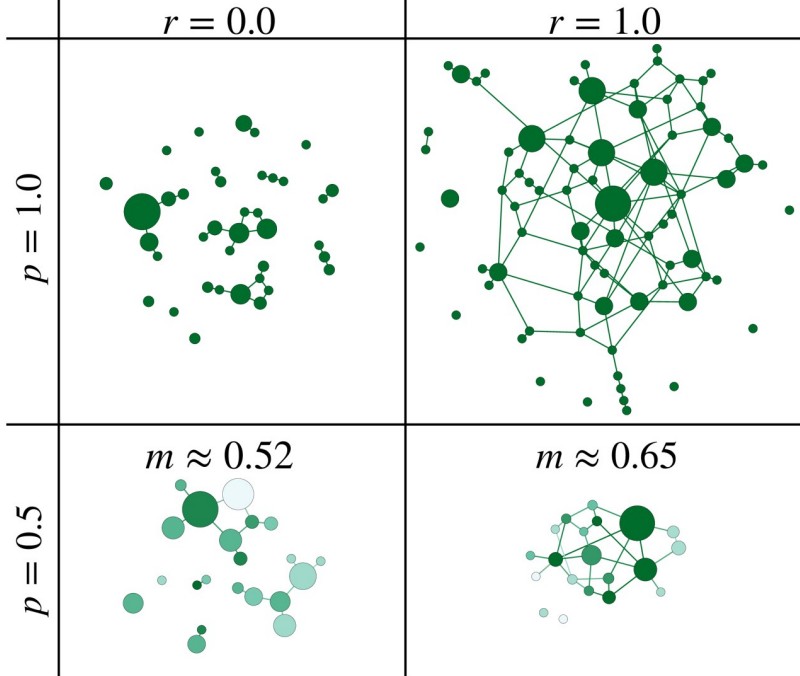

**Fig 13. Graph representation of genotype clouds in the *fsm* with $N = 100$, $L = 10$, $\mu = 0.01$.** The population has evolved for $10^6$ generations starting from a random viable genotype. Links connect genotypes that differ by a single mutation, and node sizes represent the frequency of the corresponding genotype in the population; see Illustration of results for details. The networks on the left show non-recombining populations ($r = 0$) and on the right obligately recombining populations ($r = 1$). In the top panels all genotypes are viable ($p = 1$), whereas in the bottom panels half of the genotypes are lethal ($p = 0.5$). In the bottom row the mutational robustness of genotypes is shown by color coding with dark green representing high robustness and pale green low robustness, and the average robustness $m$ is also indicated. Node sizes are normalised such that the smallest and largest node of each panel are equal in size.

evolutionary regime. The increased population size allows us to study the frequency distribution of genotypes in the population sorted by their rank. Remarkably, in non-recombining populations ($r = 0$) the distribution is exponential whereas for $r = 1$ we observe a heavy-tailed power law distribution, a feature that appears to be independent of $p$. The histograms in S11 Fig also highlight the fact that, depending on the fraction $p$ of viable genotypes, recombination can either increase or decrease the genetic diversity. In terms of mutational robustness, the graph representation shows that genotypes with high frequency exhibit an above-average robustness, thereby increasing the mutational robustness of the population.

**Time evolution.** So far we have studied the impact of recombination on stationary populations, where the effects of mutation, selection and drift balance on average. However, such a stationary state is generally not reached within a few generations, and in particular in the context of evolution experiments it is also important to understand the transient behavior. Since experiments usually consider a predefined small set of loci and track their evolution, we consider the temporal evolution in the *fsm*.

As an example, Fig 14 shows the time evolution of mutational robustness $m$ for obligately recombining and non-recombining populations at different mutation rates. The population is initially monomorphic and starts on a random viable genotype. To account for the variability between the trajectories observed in different realizations of the evolutionary process, the shading around the lines showing the average robustness represents the standard deviation of $m$. Analogous results for measures of evolvability and diversity are shown in S12–S15 Figs.

As expected, the time scale for the establishment of the stationary regime is determined primarily by the mutation rate, since mutations create the diversity on which selection and recombination can act. At the lowest mutation rate recombining and non-recombining populations behave in the same way, and with increasing mutation rate the evolutionary regimes described above are traversed. This implies in particular that the ordering between the lines representing $r = 1$ and $r = 0$ may change as a function of $\mu$ (Fig 14, S12 and S15 Figs). The distinction between recombining and non-recombining populations is often most pronounced at intermediate values of the mutation rate (S14 and S15 Figs).

## Recombination-induced genetic drift

If recombination does not occur concurrently with selection but successively, it can act as an additional source of genetic drift. This recombination-induced drift is a confounding factor that needs to be accounted for when interpreting the results obtained with successive recombination models. As an example, Fig 15 shows results for the numbers of distinct genotypes and segregating mutations for the *ism* with $p = 1$ using the simple successive recombination scheme. While in the concurrent recombination model the number of distinct genotypes is strictly increasing with $r$ at $p = 1$ (Fig 4), the effect of recombination in the simple successive model is mutation rate dependent and can be non-monotonic even at $p = 1$.

This is due to a decrease in the number of segregating mutations and the mean Hamming distance with increasing $r$ which occurs through the additional genetic drift in the simple successive recombination scheme (see S1 Appendix). Qualitatively, it can be attributed to the fact that this scheme involves two sampling events per generation, whereas the other two schemes involve only one (Fig 1). In the *fsm* the effect is similar and can result in an intermediate peak in the mutational robustness as a function of $r$ when $NL\mu \approx 1$ (S16 Fig). While the effect in the *ism* occurs at all population sizes, in the *fsm* it is only significant for relatively small populations, because the number of segregating mutations is capped at $L$.

If the design of a simulation model or an *in vitro* experiment requires that recombination and selection occur independently, but recombination should not be an additional source of

 

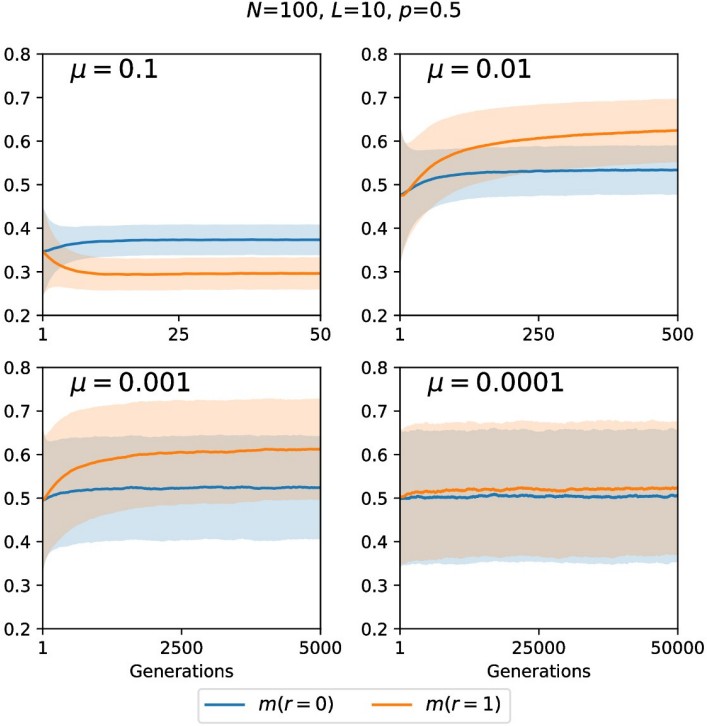

**Fig 14. Time evolution of mutational robustness in the *fsm*.** Parameters are $N = 100$, $L = 10$, $p = 0.5$ for different values of the mutation rate $\mu$. Each panel compares obligately recombining ($r = 1$) and non-recombining ($r = 0$) populations. Thick lines represent the mean over 5000 landscape realizations and the shaded areas the corresponding standard deviation.

genetic drift, then successive recombination with mating pairs, as illustrated in Fig 1, might be an option. Alternatively, one can mitigate the additional genetic drift in the simple successive recombination model by performing the selection and recombination step in each generation within a large population from which a small sample of size $N$ is subsequently drawn.

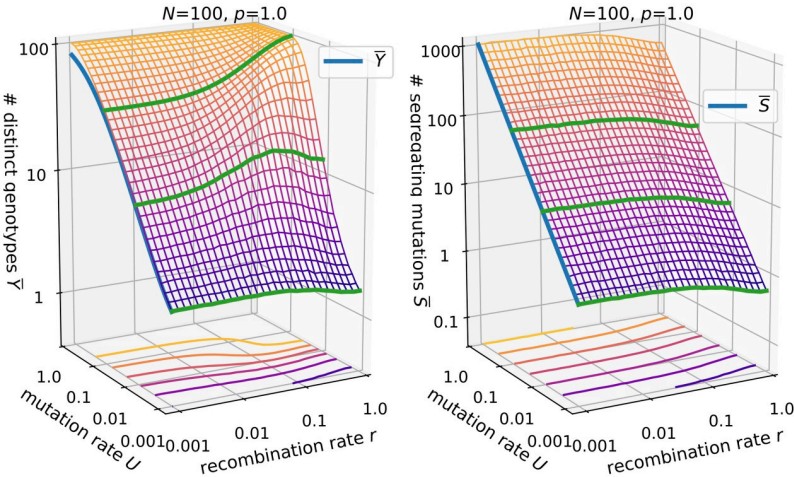

**Fig 15. Number of distinct genotypes and segregating mutations in the simple successive recombination model for the *ism*.** Parameters are $N = 100$, $p = 1$. Compare to Figs 4 and 5. Green lines show constant $U = 0.1$, $U = 0.01$ and $U = 0.001$.

## Discussion

Population genetic theories for the evolutionary benefit of recombination in the tradition of Weismann's hypothesis are based on the idea that recombination changes the evolvability of a population through its effect on the amount of useful genetic variation [29, 34–36]. Depending on additional factors such as the correlations between fitness effects of different alleles (epistasis) and their association within the population (linkage disequilibrium), this may lead to a fitness advantage of recombining populations and the selection for modifier alleles that increase the recombination rate. In this work we have investigated the interplay of recombination, evolvability and fitness for a specific class of epistatic fitness landscapes comprising extended neutral networks interspersed with low fitness genotypes. In fully connected networks we find a simple positive correlation between the recombination rate and measures of evolvability, but the behavior in the presence of unfit genotypes is much more complex.

When unfit genotypes are randomly distributed in sequence space and if their fraction is large enough, we find the emergence of two different regimes for the effect of recombination. While small recombination rates increase diversity in accordance with Weismann's hypothesis, at sufficiently high recombination rates we observe a strong contraction of the genotype cloud and reduced evolvability. This contraction regime is characterized by a clustering of the population in sequence space around a focal genotype, which we have shown to be the most recent common ancestor. Therefore most genotypes have only a few segregating mutations as well as a small pairwise mean Hamming distance, which leads to a reduced number of distinct genotypes. This structural change in genotype composition of the population at sufficiently high recombination rates leads to a recovery in the average fitness, as recombination events are more likely to result in viable genotypes.

Our results for the finite-sites model further reveal, that polymorphic genotype clouds are most dense around mutationally robust genotypes and that a contraction through recombination therefore greatly increases mutational robustness. The trade-off is a reduced evolvability in terms of the number of generations until all viable genotypes are discovered. Strongly recombining populations can even become trapped in mutationally robust regions, such that the time to full discovery diverges. The recombination-induced trapping of the population at fitness peaks is well known from studies on non-neutral fitness landscapes [39, 66–70], but our results show that a similar phenomenon occurs in neutral landscapes with fitness plateaus and ridges.

In the infinite-sites model reduced evolvability at high recombination rates is manifested through a reduced discovery rate. Furthermore, the lower frequency of segregating mutations leads to a significantly reduced fixation rate. Therefore, even in an infinite-sites setting, recombination can, in some sense, entrench the population. However, as the number of potential mutation sites is unbounded, the discovery rate never falls below its value in the absence of recombination. In terms of genetic diversity the results in the infinite and finite-sites settings are similar but more gradual in the latter case, as the number of segregating mutations and the mean Hamming distance are capped.

Overall, our numerical simulations show a very consistent increase in mutational robustness with the recombination rate in polymorphic populations. Related to this is the observation that recombination leads to a heavy-tailed frequency distribution of genotype abundance, and that the most frequent genotypes have an above-average robustness, thereby increasing the robustness of the whole population. As discussed in previous work, the most frequent genotypes are more likely those that have an above-average fraction of possible viable parent combinations [26].

Consistent with the earlier analytic results obtained for infinite populations [26], our simulations indicate that selection for robustness and reduction of evolvability set in once recombination events become more common than mutation events (see e.g. S2 Fig and S3 Fig). Furthermore, this finding is independent of the exact nature of the recombination event (S5 Fig and S6 Fig). In other words, the recombination rate $r$ does not need to be large in absolute terms, but merely exceed the genome-wide mutation rate $U$. This observation is relevant for the applicability of our results to facultative sexual organisms, which typically display values of $r/U$ that are greater than one even when sex and recombination is very rare [71–74]. We therefore expect that recombination can contribute to reducing the genetic diversity of real populations also in such cases.

In the interest of conceptual simplicity we have restricted our study to percolation landscapes in which the unfit genotypes are randomly distributed in sequence space. An important question for future work is to generalize our findings to neutral network structures arising from realistic genotype-phenotype maps [5, 52, 75]. Moreover, in our study, we neglected mutations with small deleterious fitness effects. These are also relevant for the evolution of recombination, as populations with increased recombination rates are more efficient in purging slightly deleterious mutations [48].

The broad scope of our investigation demonstrates that the effects of recombination vary widely across parameter combinations and evolutionary regimes, and this has to be accounted for when interpreting apparent contradictions between different experiments. Furthermore, it is important to distinguish long-term effects of recombination from short-term effects. In this study, we mainly considered long-term effects in stationary populations. Short-term effects can be different, in particular when evolution proceeds in a changing environment [40, 76].

## Supporting information

**S1 Appendix. This appendix contains the analytic derivation of the mean Hamming distance for different recombination schemes.**
(PDF)

**S1 Fig. Probability that the network of viable genotypes is connected in a percolation landscape.** Genotypes are viable or lethal independently with probability $p$. The sequence length is $L = 10$.
(PDF)

**S2 Fig. Supplementary information to Fig 9.** The right column with $U = 0.1$ is identical to that of Fig 9, and the left column illustrates the behavior at mutation rate $U = 0.05$. The results show that at lower mutation rates the contraction of the genotype cloud occurs already at larger fractions of viable genotypes ($p = 0.7$) and moreover shifts to smaller recombination rates. At the same time the magnitude of the effect of recombination becomes smaller for smaller $U$. At even lower mutation rates the population becomes monomorphic and recombination has no effect.
(PDF)

**S3 Fig. Discovery rate in the *ism* for two different population sizes.** The population size is $N = 1000$ (left panel) vs. $N = 100$ (right panel) with $p = 0.5$. The green line is drawn at $U = 0.1$ and the blue lines in both panels show Eq 11. The right panel is identical to that in Fig 2.
(PDF)

**S4 Fig. Varying fitness of unfit genotypes.** The figure shows the results of simulations for $U = 0.05$ and $p = 0.7$ in which the fitness of the unfit genotypes is varied between $w_0 = 0$ and $w_0$

= 0.6, whereas high fitness genotypes have fitness $w_1 = 1$. The contraction of the genotype cloud at large recombination rates is seen to persist up to $w_0 = 0.4$ for the chosen parameters, showing that lethality of the deleterious mutations is not a necessary condition for the results presented in the main text.
(PDF)

**S5 Fig. One-point crossover vs. uniform crossover.** Dependence of evolvability measures on recombination rate obtained with the uniform crossover scheme employed in the main text is compared to results obtained with a one-point crossover scheme. The right column with $U = 0.1$ is identical to that of Fig 9. In order to implement the one-point crossover in the *ism*, each novel mutation is assigned a genomic position as a uniform random variable within the range [0, 1], and the position of the crossover is determined by another random number within this range. The non-monotonic behavior persists, but the measures vary more slowly with $r$. This is likely due to the fact that the single-point crossover produces less diversity compared to the uniform crossover.
(PDF)

**S6 Fig. Obligate vs. facultative sexual reproduction.** The right column with $U = 0.1$ is identical to that in Fig 9 and S5 Fig. Whereas in the main text the recombination rate $r$ describes the recombining fraction of the population, in the left column all individuals recombine and $r$ describes instead the average number of crossovers, which is assumed to be Poisson distributed. Crossovers are implemented as described in the caption of S5 Fig, with the additional feature that multiple crossovers can occur. The results show that a non-monotonic dependence of $r$ can still arise but, similar to S5 Fig, the variation with $r$ occurs more slowly.
(PDF)

**S7 Fig. Mean fitness and viable recombination fraction in the *fsm*.** Parameters are $N = 100$, $L = 10$, $p = 0.5$. The green line is drawn at $\mu = 0.01$ in both panels. Similar to the results for the *ism* (Fig 8), the fitness displays an intermediate minimum at the point where the population structure changes. This is best visible at $\mu = 0.03$ (blue line). Compared to the *ism* the variation in mean fitness and viable recombination fraction is less pronounced.
(PDF)

**S8 Fig. Discovery rate of an escape variant.** Here the time is measured until a randomly chosen escape variant is found by at least one individual in the *fsm*. Top panel shows the inverse discovery time averaged over all simulations, as well as the inverse time for a subset of simulations where the escape variant has either robustness $m_{escape} = 1.0$ or $m_{escape} = 0.6$. These subsets represent scenarios where the robustness of the escape variant is either larger or lower than the average robustness $p = 0.8$. The results show that more robust genotypes are detected more quickly and less robust genotypes less quickly. The bottom panel shows analogous results where the subsets are distinguished according to the robustness of the initial genotype. In this case the discovery time is not affected.
(PDF)

**S9 Fig. Reciprocal of the total number of mutation events until full discovery in the *fsm*.** Parameters are $N = 100$, $L = 10$ and $p = 1.0$ (left panel) vs. $p = 0.5$ (right panel). The green line is drawn at $L\mu = 0.1$.
(PDF)

**S10 Fig. Genetic diversity in the *fsm*.** Parameters are $N = 100$, $L = 10$ and $p = 1.0$ (left column) vs. $p = 0.5$ (right column). The green line is drawn at $L\mu = 0.1$. The blue line in the bottom left

panel shows the expression in Eq 20 for the mean Hamming distance at $p = 1$.
(PDF)

**S11 Fig. Graph representation of genotype clouds in the *fsm* with $N = 10000$, $L = 14$, $\mu = 0.0001$.** Supplementary information to Fig 13. The population has evolved for $10^7$ generations. Inset histograms show the frequency distribution sorted by rank. The histograms are in semi-log scale for $r = 0$ and in log-log scale for $r - 1$. Through the number of ranks, the histograms also display the number of existing distinct genotypes. Note that for both values of $p$ recombination makes the frequency distribution heavy-tailed, but it may either increase or decrease the number of distinct genotypes. Except in the upper right panel ($r = 1$, $p = 1$), the node sizes are normalized such that the smallest and largest nodes in each panel have the same size.
(PDF)

**S12 Fig. Time evolution of the number of distinct genotypes in the *fsm*.** Each panel compares obligately recombining ($r = 1$) and non-recombining ($r = 0$) populations. Thick lines represent the mean over 5000 landscape realizations and the shaded areas the corresponding standard deviation.
(PDF)

**S13 Fig. Time evolution of the number of segregating mutations in the *fsm*.** Parameters are $N = 100$, $L = 10$, $p = 0.5$ with four different values of the mutation rate $\mu$. Each panel compares obligately recombining ($r = 1$) and non-recombining ($r = 0$) populations. Thick lines represent the mean over 5000 landscape realizations and the shaded areas the corresponding standard deviation.
(PDF)

**S14 Fig. Time evolution of the pairwise mean Hamming distance *fsm*.** Parameters are $N = 100$, $L = 10$, $p = 0.5$ with four different values of the mutation rate $\mu$. Each panel compares obligately recombining ($r = 1$) and non-recombining ($r = 0$) populations. Thick lines represent the mean over 5000 landscape realizations and the shaded areas the corresponding standard deviation.
(PDF)

**S15 Fig. Time evolution of the fraction of explored viable genotypes in the *fsm*.** Parameters are $N = 100$, $L = 10$, $p = 0.5$ with four different values of the mutation rate $\mu$. Each panel compares obligately recombining ($r = 1$) and non-recombining ($r = 0$) populations. Thick lines represent the mean over 5000 landscape realizations and the shaded areas the corresponding standard deviation.
(PDF)

**S16 Fig. Mutational robustness in the *fsm* and simple successive recombination dynamics.** Parameters are $N = 100$, $L = 10$, $p = 0.5$. The green line at $\mu = 0.001$ ($NL\mu = 1$) shows a non-monotonic variation with recombination rate, which is caused by recombination-dependent genetic drift.
(PDF)

## Acknowledgments

We thank Thomas Wiehe for helpful comments on the manuscript.

## Author Contributions

**Conceptualization:** Alexander Klug, Joachim Krug.

**Formal analysis:** Alexander Klug.

**Investigation:** Alexander Klug, Joachim Krug.

**Methodology:** Alexander Klug.

**Project administration:** Joachim Krug.

**Software:** Alexander Klug.

**Supervision:** Joachim Krug.

**Visualization:** Alexander Klug.

**Writing – original draft:** Alexander Klug.

**Writing – review & editing:** Alexander Klug, Joachim Krug.

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
