## [Decision Letter · Decision Letter 0]

19 Sep 2022

Dear Dr. Krug,

Thank you very much for submitting your manuscript "Conflicting effects of recombination on the evolvability and robustness in neutrally evolving populations" for consideration at PLOS Computational Biology. As with all papers reviewed by the journal, your manuscript was reviewed by members of the editorial board and by several independent reviewers. The reviewers appreciated the attention to an important topic. Based on the reviews, we are likely to accept this manuscript for publication, providing that you modify the manuscript according to the review recommendations. Both referees found your work to be of considerable interest and scientifically sound. One referee would like you to address how this work relates to real organisms, at least in the discussion section.

Sincerely,

Sergei Maslov

Academic Editor

PLOS Computational Biology

Natalia Komarova

Section Editor

PLOS Computational Biology

[LINK]

Both referees found your work to be of considerable interest and scientifically sound. One referee would like you to address how this work relates to real organisms, at least in the discussion section.

Reviewer's Responses to Questions

**Comments to the Authors:**

Reviewer #1: In their manuscript, Klug and Krug study the role of recombination in modulating mutational robustness, discovery, and evolvability. They study this effect for finite populations with finite number of alleles, to differentiate their work from standard population genetic models.

While traditionally, it is assumed that recombination allows explorations in the genetic space, thereby increasing population diversity and evolvability, the authors show that at very high recombination rates and especially in a finite site model, recombination can homogenize the population around mutationally robust genotypes which increases the robustness of the population but at the cost of evolvability and diversity. The work is sound, well written, and well supported by a plethora of supplementary figures. Overall, this is an interesting study and deserves publication in a theoretical/computational biology journal.

However, I have one major concern. Currently, there is no real discussion on how their results map onto real systems. I think it will greatly benefit from some connection to reality. For example, the populations they talk about are facultative sexual (for example, yeast). Could they compare the parameters and dimensionless numbers in their simulations with real life evolutionary parameters of facultatively sexual organisms? Right now, it is not clear whether the recombination rates where the tradeoff is observed are too high for realistic populations. If that is the case, then the study becomes less interesting as it is exploring parameter regimes that are not realistic.

In short, I would like the authors to address how their work relates to real organisms, at least in the discussion section.

Reviewer #2: Klug and Krug (KK) investigate the effect of recombination on evolvability, genetic variation, and mutational robustness in populations evolving over neutral networks (NN), also known as holey fitness landscapes. Genotypes in the NNs either have a fitness of 1 with probability p or are lethal otherwise. The fitnesses of genotypes are assigned at random.

Broadly, KK consider two models: infinite- and finite-sites models (ISM and FSM). The two models differ in subtle ways such as that there are no back mutations in the ISM. KK compare the outcome of evolution on fully connected NNs without lethal genotypes (p=1) and on NNs with lethal genotypes (typically, p=0.5). They evaluate several metrics of evolvability, genetic variation, and mutational robustness in populations evolving under different mutation and recombination rates. Most results presented were obtained through stochastic simulations but they also derived several major analytical results.

I believe this work fills an important gap in the NN literature. Most investigations of evolution on NNs have only considered asexual reproduction. The methods are sound and the paper is clearly written. Therefore, I am supportive of publication in PLoS Computational Biology.

**Have the authors made all data and (if applicable) computational code underlying the findings in their manuscript fully available?**

Reviewer #1: **No: **The authors note that the codes "will be uploaded" on github but I couldn't find a link.

Reviewer #2: **No: **I couldn't find any reference to data and code availability. Maybe I missed it.

PLOS authors have the option to publish the peer review history of their article (what does this mean?). If published, this will include your full peer review and any attached files.

Reviewer #1: No

Reviewer #2: No

Figure Files:

Data Requirements:

Reproducibility:

References:

---

## [Editor Report · Decision Letter 1]

4 Nov 2022

Dear Dr. Krug,

We are pleased to inform you that your manuscript 'Conflicting effects of recombination on the evolvability and robustness in neutrally evolving populations' has been provisionally accepted for publication in PLOS Computational Biology.

Best regards,

Sergei Maslov

Academic Editor

PLOS Computational Biology

Natalia Komarova

Section Editor

PLOS Computational Biology

---

## [Editor Report · Acceptance letter]

17 Nov 2022

PCOMPBIOL-D-22-01022R1 

Conflicting effects of recombination on the evolvability and robustness in neutrally evolving populations

Dear Dr Krug,

I am pleased to inform you that your manuscript has been formally accepted for publication in PLOS Computational Biology. Your manuscript is now with our production department and you will be notified of the publication date in due course.

With kind regards,

Zsofi Zombor
